# Generalizing Verifiable Instruction Following

**Valentina Pyatkin**[αβ]    **Saumya Malik**[α∗]    **Victoria Graf**[αβ∗]

**Hamish Ivison**[αβ] **Shengyi Huang**[α] **Pradeep Dasigi**[α] **Nathan Lambert**[α] **Hannaneh Hajishirzi**[αβ]

[α]Allen Institute for Artificial Intelligence
[β]University of Washington
contact: valentinap@allenai.org

## Abstract

A crucial factor for successful human and AI interaction is the ability of language models or chatbots to follow human instructions precisely. A common feature of instructions are output constraints like "only answer with yes or no" or "mention the word 'abracadabra' at least 3 times" that the user adds to craft a more useful answer. Even today's strongest models struggle with fulfilling such constraints. We find that most models strongly overfit on a small set of verifiable constraints from the benchmarks that test these abilities, a skill called precise instruction following, and are not able to generalize well to unseen output constraints. We introduce a new benchmark, IFBENCH, to evaluate precise instruction following generalization on 58 new, diverse, and challenging verifiable out-of-domain constraints. In addition, we perform an extensive analysis of how and on what data models can be trained to improve precise instruction following generalization. Specifically, we carefully design constraint verification modules and show that reinforcement learning with verifiable rewards (RLVR) significantly improves instruction following. In addition to IFBENCH, we release 29 additional new hand-annotated training constraints and verification functions, RLVR training prompts, and code.

## 1 Introduction

Following instructions *exactly* is a crucial skill for a language model to have, in order for it to generate a useful output that corresponds to the entirety of a user's specifications and not just the general topic. In particular, instructions often include output constraints that specify length, format and content. A model's ability to follow constraints in instructions is evaluated on precise instruction following benchmarks with verifiable constraints, with IFEval [36] being the most popular benchmark (and it has quickly saturated, with many leading models scoring 80+% at as small as 2B parameters [5, 15, 25].[2] This evaluation benchmark consists of a set of 25 constraint templates, which can all be automatically verified using short python functions. Most models strongly overfit to this small set of constraints.

In order to investigate a model's precise instruction following (IF) generalization abilities, we introduce IFBENCH with new, diverse, and challenging verifiable instruction following constraints where leading models such as GPT-4.1 or Claude 3.7 Sonnet score below 50%. The constraints in IFBENCH cover important skills like counting, formatting, sentence/word/character manipulations, and copying. This shows that most state-of-the-art models overfit on IFEval and are not able to generalize well to the unseen constraints we introduce. Figure 1 shows the discrepancy in accuracy between IFEval and IFBENCH for state-of-the-art models.

---

[∗]Joint second authors.

[2]Technically not every constraint passed by users is *simple* to verify with software, but these evaluations focus on implementable variants for ease of iteration and improvement.

39th Conference on Neural Information Processing Systems (NeurIPS 2025) Track on Datasets and Benchmarks.

To facilitate experimenting with the generalization of precise IF, we create 29 training constraints by manually curating useful and challenging constraints and verification functions, some of which are representative of real-world chatbot usage, while others are inspired by the core capabilities we desire our models to have. We find that increasing the number and variety of training constraints improves IF generalization. In addition to creating distinct training constraints, we explore new methods for inducing strong IF performance by appending combinations of constraints and using wider constraint variable ranges for the training prompts in the RL stage, rewarding models for following more complex instructions. For example, an instruction could be combined with a length constraint, a formatting constraint and a constraint asking to include specific words.

Given that many precise instruction following constraints are verifiable, we show how to use novel reinforcement learning with verifiable reward (RLVR) techniques to train models to be better at precise IF while maintaining performance on existing skills [15, 8]. To gain further insights into when generalization happens for precise instruction following, we analyze the effect of training data and post-training algorithms on IF performance, both in and out-of-domain. Our results indicate that RLVR with Group Region Policy Optimization (GRPO) [21] and data augmentation leads to significant performance increases on old IF benchmarks and IFBENCH.

Beyond improving precise IF, we also see that RLVR trained models exhibit different instruction following behaviors compared to their non reinforcement trained counterparts. Take for example an instruction like "write a recipe for tiramisu" with an output constraint like *only use unique words in your output, do not mention any word more than once*. There is a tension between following the task of writing a recipe and adhering to the constraint. Many models would prioritize the task, while IF-RLVR trained models tend to prioritize the constraint – future work will explore how to blend these behaviors with refined training recipes. Qwen2.5-72B-Instruct, for example, tends to prioritize the constraint over the instruction. In line with the above example of constraint-focused generation, we find that RLVR can lead to over-optimization and we suggest adding a preference reward model signal to balance the signals. Our contributions are as follows:

1. A new, unseen and challenging precise instruction following benchmark, IFBENCH[3], with 58 new constraints and corresponding verification functions. With an investigation into the generalization abilities of LLMs for following constraints, we find that leading LMs such as Qwen3-32B or Claude 4 Sonnet score below 50% showcasing the opportunity for improvement in precise IF.

2. 29 new training constraints and verification functions, IFTRAIN, to enable simple data creation that improves instruction following performance.

3. New methods of RLVR training for precise instruction following (IF-RLVR) by interleaving multiple constraints per prompt or mixing verifiable and preference rewards. With our new training techniques we improve the IFeval scores of a TÜLU-3-8B model from 82.4 to 92.2, and the IFBENCH scores from 28.9 to 45.9. Similarly, IF-RLVR increases a Qwen-2.5-7B base model to scores of 87.8 (IFEval) and 54.7 (IFBENCH), and we also show that our approach is effective for models from the OLMo family.

## 2 IFBENCH & IFTRAIN: Measuring and Training Precise IF

In this section we specify the problem specification of precise instruction following so that we can detail the benchmark construction process and its final contents. Along with introducing IFBENCH, we detail how we can use similar methods to create IFTRAIN and train models that generalize better to a broad range of instruction following tasks.

The task of precise instruction following (IF) evaluates a language model's ability to perform a task $t$, such as summarization or creative writing, while adhering to one or more output constraints $c$, which can be automatically verified. Users naturally use constraints in their prompts [34, 16], so precise IF is an important task to master and most models report performance scores on IFEval. Many models even have designated sections in their technical reports on how they improve IF performance – the most common approach is targeted synthetic data generation [32, 15, 1, 7]. The Nemotron-4 340B technical report [1] goes into more detail and mentions that targeted synthetic IF data is generated by combining synthetic instructions with constraints from the IFEval taxonomy. Figure 1 shows that

---

[3]Code for IFBENCH is available here: `https://github.com/allenai/IFBench`.

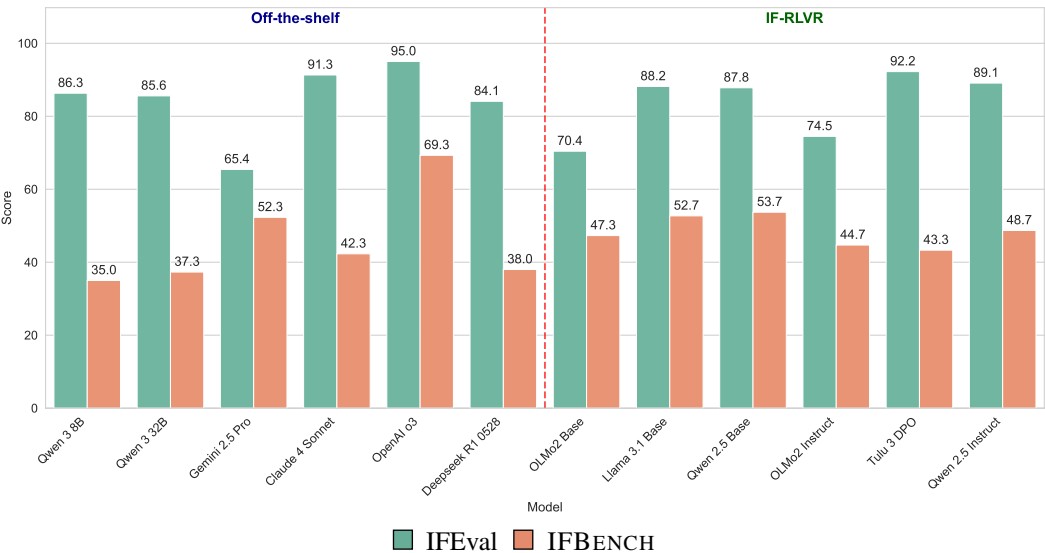

Figure 1: Model performance on IFEval and IFBENCH (single-turn). Left Models: out-of-the-box performance. Right Models: after IF-RLVR training. IFBENCH has either 1 or 2 constraints per instruction.

models display good accuracy on IFEval. The scores on our new unseen benchmark, IFBENCH, on the other hand, are much lower due to the verifiable constraints being different, despite the task and evaluation setup being the same. This discrepancy between the results indicates that most models overfit to a small set of verifiable constraints and do not possess sufficient capabilities to generalize as well to unseen constraints.

We introduce a new benchmark, IFBENCH, and paradigm to evaluate the generalizability of methods addressing precise instruction following. We define a taxonomy of constraint templates, which we split into training and test constraints to prevent contamination. The new constraints we introduce were created manually – sourced by collecting feedback from LM users beyond the authors on the types of constraints they have tried with models, or manually written to cover core IF skills. Then, we filtered constraints for the benchmark to those that can be easily paired with a verification function written in Python, making for reproducible evaluation and training tools. The full list of new constraints can be found in Appendix A for evaluation and Appendix B for training constraints, with an analysis in Appendix D.

**IFBENCH** consists of 58 new verifiable constraints that go beyond the 25 constraints included in IFEval [36]. To create the final test prompts, we add instantiated constraints to unseen, i.e. held out from release, prompts from WildChat [34]. By combining unseen prompts with unseen constraints, we prevent accidental train-test contamination and can appropriately evaluate language models' abilities to generalize on the task of precise instruction following. Every instance went through a human annotation process to verify the prompt and constraint compatibility (i.e. a coding related constraint, for example, does not fit with a prompt asking for a summary). These constraints cover 7 different broader categories: *count, ratio, words, sentence, format, custom, copy.* These categories cover a broad range of sub-skills, such as a model's ability to copy parts of the input prompt into the output. The final benchmark consists of 300 prompts. When curating the benchmark, another focus was to include challenging constraints, such as *Maintain a 2:1 ratio of declarative to interrogative sentences.* Similarly to the original IFEval, we compute both strict and loose accuracy, where the strict accuracy verifies if the constraint is followed correctly, and the loose accuracy additionally cleans the model's output by removing first/last lines and certain font modifiers. Each instruction in the benchmark contains either one or two constraints. We evaluate constraint following abilities in two different settings:

- Single-turn: The "user" prompt consist of a general prompt with task $t$, concatenated with one or more output constraints $c$, and the model has to complete $t$ while adhering to $c$.

- Multi-turn: $c$ is isolated from $t$ in three turns. The first "user" prompt consist of a general prompt with task $t$ and the second turn is an "assistant"'s response to $t$, $r_1$. The third turn ("user") asks to rewrite $r_1$ to comply with a constraint $c$. The model has to respond to the third turn, given all previous turns as context.

**IFTRAIN** consists of 29 new, unseen, verifiable constraints, with their corresponding verification functions. This more than doubles the current set of train constraint types. The full list of new constraints can be found in Appendix B. The constraints were created to capture the basic building blocks of classic constraints. For example, to teach the model to copy better from the input, we create different versions of copying tasks, such as copying spans or copying and editing of the input.

## 3 IF-RLVR

In what follows we expand upon a new approach for training language models on precise instruction following with reinforcement learning with verifiable rewards [15], IF-RLVR. We propose the following training and data recipe to achieve strong in and out-of-domain IF performance.

**Data:** We create targeted IF-RLVR training data that is diverse and covers a variety of constraints. The prompts for verifiable IF training are created by combining an instruction from a public SFT dataset with a constraint from either the IFEval taxonomy (under Apache 2.0 license) or our new unseen training constraint taxonomy (which is separate from the constraints in IFBENCH). We randomly sample prompts from TÜLU-3-SFT [15] and we append at least one and up to $n$ constraints. We prevent the combination of contradictory constraints by maintaining a dictionary of constraint conflicts. As training constraints we use IFTrain and the constraints from IFEval, which we expand by increasing the variable range for each constraint. For most of the experiments we create about 60k-100k prompts.

**Training** Reinforcement Learning with Verifiable Rewards (RLVR) [15] can be applied to the precise instruction following task, as each constraint can be verified with a function. Specifically, we train a policy with GRPO [21] and outcome supervision, where each output is scored according to wether or not the constraint has been correctly fulfilled.

For multi-constraint IF-RLVR, the reward per instance is calculated as follows:

$$\text{Instance Reward} = \sum_{i=1}^{n} \text{verifiable\_reward}_i \cdot \text{reward\_multiplier}_i \cdot \text{reward\_weight}_i \tag{1}$$

Where the reward multiplier and the reward weights are hyperparameters, generally set to 1, except if one wants to up or down-weight certain rewards. IF-RLVR training works for base models, using a special chat template (see Appendix F), and for post-trained models, using their own chat templates.

**Experimental & Evaluation Setup** We experiment with the following base policies: Llama-3.1-Tulu-3-8B-DPO, Llama-3.1-8B, Qwen2.5-7B, Qwen2.5-7B-instruct, OLMo2, OLMo2-instruct. We train using the GRPO implementation in open-instruct [11, 15], with the following hyperparameters: max_token_length=2048, temperature=1, learning rate=5e-7, 16 samples per prompt, 8 GPUs (H100) and a local mini-batch size of 32. Training took on average 1 day for 2000 steps. Base models with a reasoning chat template are trained with a max token length of 10240, a beta of 0, and a temperature of 1. We report the prompt-level loose accuracy for both IFEval and IFBENCH. See App. G for more evaluation details.

### 3.1 IF-RLVR Results

In Figure 1 we show that IF-RLVR is very well suited to for teaching a model to follow instructions precisely. Our IF-RLVR trained models outperform most of the current state-of-the-art models (besides o3). We also show that our recipe can be successfully applied to 3 different model families: OLMo [18], Qwen 2.5 [26], and Llama 3.1 [27].

Table 1: Training on 1-6 constraints per instance. Training on 50-1000 instances per constraint. (qwen2.5 policy)

|  | 1 | 2 | 3 | 4 | 5 | 6 | 10 | 50 | 100 | 500 | 1000 |
|---|---|---|---|---|---|---|---|---|---|---|---|
| IFBench | 48.9 | 53.1 | 59.5 | 49.4 | 55.8 | 54.1 | 48.6 | 52.7 | 51.7 | 51 | 48.6 |
| IFEval | 71.2 | 79.9 | 77.8 | 79.5 | 79.9 | 85.8 | 73.6 | 72.8 | 74.3 | 70.1 | 72.8 |

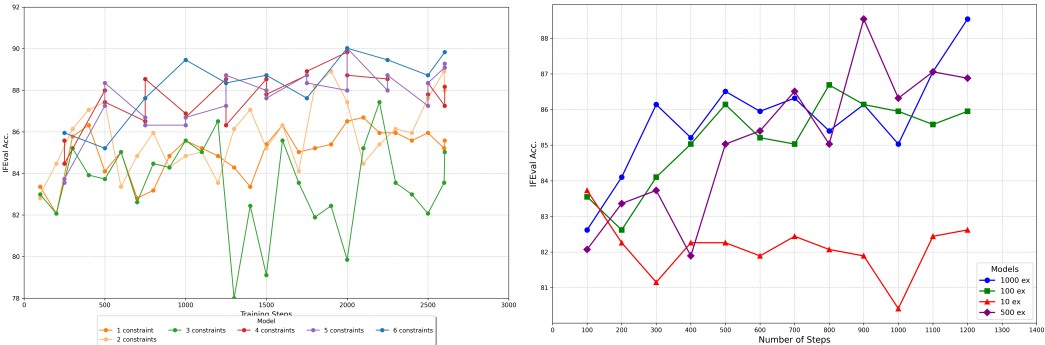

Figure 2: Training on 1 - 6 constraints per instruction. (TÜLU-DPO policy)

Figure 3: Training on 10, 100, 500 and 1000 instances per constraint.

## 4  IF-RLVR Experiments

In this section we ablate our modeling design and data choices, and compare IF-RLVR to other training approaches like DPO.

### 4.1  Training on Multiple Constraints

We experiment with RL training on multiple constraints per instance. For each instruction $i$, randomly sampled from TÜLU-SFT [15], we append at least one and up to $n$ constraints, where $n \in \{1, 2, 3, 4, 5, 6\}$. This results in four different sets of RLVR training data with multiple constraints. We prevent the combination of contradictory constraints by maintaining a dictionary of constraint conflicts.

We find that **training on a combination of constraints improves both in-domain and out-of-domain performance.** As displayed in Figure 2, training on a bigger combination of constraints leads to better performance, compared to training on only up to 3 constraints per instance. Interestingly, instructions in IFEval have up to 3 constraints and up to 2 constraints in IFBENCH, but training on up to 5 or 6 constraints still leads to better generalization on these benchmarks. Also on the out-of-domain benchmark IFBENCH, the best performance is achieved when training on more than one constraint per instance (Table 1).

### 4.2  Seen vs. Unseen Constraints

Training on the 25 constraint templates from IFEval directly translates to good performance on IFEval. We perform multiple GRPO training runs where we take the 29 'unseen' constraint templates from IFTRAIN, which do not overlap with any of the test constraints, and add $n$ 'seen' constraint templates (from IFEval), with $n \in \{5, 10, 15, 20, 25\}$.

A combination of the full IFTRAIN and IFEval constraints leads to the highest in-domain performance on IFEval (see Fig. 4). On the out-of-domain benchmark IFBENCH, on the other hand, performance is less affected by the number of IFEval constraints the model is trained on. Nevertheless, we see that training on a larger set and larger variety of constraints is beneficial for generalization.

### 4.3  Changing the range of constraint variables

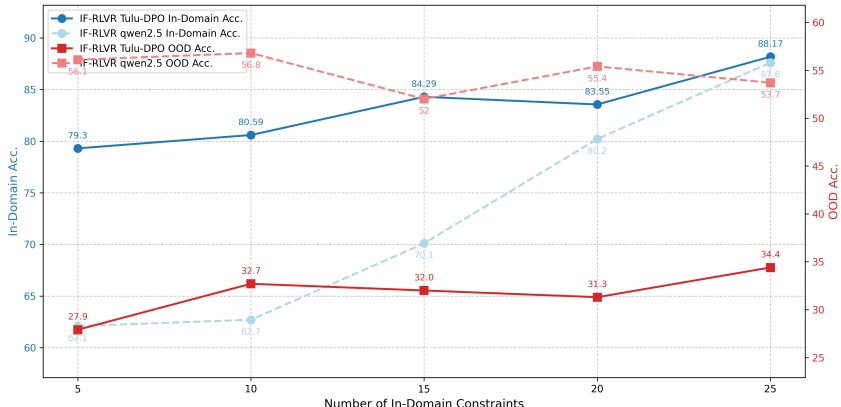

Figure 4: Training on IFTrain (ood) + n constraints (in-domain) from IFEval. (TÜLU-DPO policy and Qwen2.5)

Most of the constraint templates contain variables which can be instantiated with different values. *In your response, all lowercase words should appear at most N times.*, for example, has the variable $N$ which could in theory be any number. For both the IFEval and the IF-BENCH benchmarks, variables are instantiated for each instruction instance from a fixed range of options. Another type of generalization to assess is whether a model trained on the same constraints, but different variable ranges, can still apply the skill to unseen ranges. To evaluate this, we chose variable ranges for training that are disjoint from the test ranges. For the constraint *Your response should contain at most num_sentences sentences.*, for example, we sample a value between 20 and 40 for train and a value between 1 and 20 for test. Specifically we

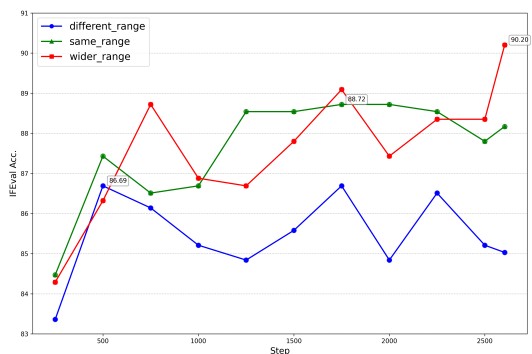

Figure 5: Experiments with variable ranges. (TÜLU-DPO policy)

experiment with three different settings: DIFFERENT RANGE, where every train variable is filled from a range that is disjoint from the test range; WIDER RANGE, where every train variable is filled from a range that includes and extends the test range; SAME RANGE, where variables are filled with the same train/test range. Fig. 5 shows performance on IFEval for different steps of IF-RLVR training. Training on a different variable range consistently scores lower than the other two setups. Though training on a wider variable range, performs comparably and often even better than training on the same range. This suggests that training on a diverse set of constraint variables improves generalization for in-domain constraints performance.

## 4.4 Removing Categories

Most verifiable constraints fall under broader constraint type categories. IFEval, for example, has 9 different categories, such as LENGTH CONSTRAINTS or DETECTABLE FORMAT. To investigate how training on a set of categories generalizes to unseen categories, we experiment with training on a leave-one-out set of categories, iteratively removing one of the following classes: CHANGE CASES, DETECTABLE FORMAT, LENGTH CONSTRAINTS, KEYWORDS.

Removing the constraints from the LENGTH and the KEYWORDS categories harms IFEval performance the most, while removing constraints from the CHANGE CASES and DETECTABLE FORMAT categories barely affect performance with the model achieving an IFEval accuracy of 89.65 (see Fig. 6).

## 4.5 Teaching the basic units of precise IF

Table 2: IFEval (blue) and IFBench (yellow) performance breakdown for constraint types, for the final IF-RLVR models and one of the base models (TÜLU-DPO), as comparison.

| | IFEval | IFB. | case | detect. | keywo. | length | count | format | words | sent. |
|---|---|---|---|---|---|---|---|---|---|---|
| tulu3DPO | 81.1 | 25.5 | 82.9 | 94.2 | 79.3 | 68.6 | 45.3 | 35.2 | 7.0 | 13.3 |
| IF-RLVR (tulu3DPO) | 92.2 | 44.6 | 92.0 | 99.2 | 95.4 | 89.3 | 53.1 | 64.3 | 48.1 | 36.7 |
| IF-RLVR (qwen2.5) | 87.8 | 53.7 | 93.3 | 94.4 | 93.2 | 90.9 | 67.7 | 68.1 | 67.0 | 71.1 |

Table 3: Model performances on different benchmarks.

| | IFEval | IFBench | Alpaca | GSM8K | MMLU | BBH |
|---|---|---|---|---|---|---|
| TÜLU-sft | 72.8 | 20.7 | 12.4 | 76.2 | 65.9 | 69.7 |
| TÜLU-DPO | 81.1 | 25.5 | 33.5 | 84.3 | 68.7 | 68.7 |
| TÜLU | 82.4 | 28.9 | 34.5 | 87.6 | 68.2 | 69.0 |
| IF-RLVR TÜLU-DPO 8b | 92.2 | 44.6 | 21.3 | 83.2 | 66.4 | 68.9 |
| IF-RLVR qwen2.5 7b | 87.8 | 53.7 | 1.1 | 15.3 | 59.4 | 26.0 |

We designed the new training constraints so that they would cover IF skills models are currently lacking in, such as copying from the input, counting, and formatting. We find that GRPO training on our new constraints shows targeted improvements in all these areas. As seen in Table 2, our final two models (from base vs. from instruct) improve over the base model (in this case a DPO trained model) in all categories, such as in counting, inserting the right amount of keywords, formatting and in following length constraints. Most of the IFEval categories seem saturated with IF-RLVR training (performance > 90), while the IFBENCH categories leave room for improvement, such as the *words* and *sentence* categories.

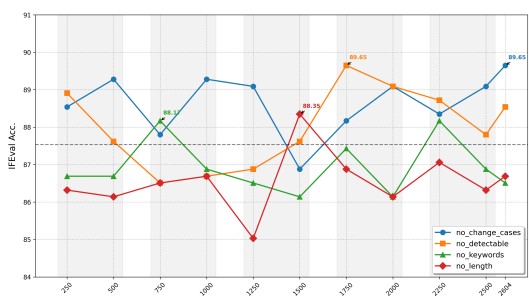

Figure 6: Removing a constraint category from training. (TÜLU-DPO policy)

Comparing our final models to other post-trained (SFT, DPO, RLVR) models in Table 3, we find that our approach and model results in the best in- and out-of-domain instruction following performance on both IFEval and IFBENCH. We also see that targeted RLVR training for IF slightly harms other downstream evaluations, such as AlpacaEval 2, while staying comparable on others, such as GSM8K, MMLU and BBH. We therefore suggest, for future work, to investigate how to combine precise IF RLVR with other types of rewards for other tasks such as math or chat. Note that the performance on other benchmarks is low for IF-RLVR Qwen2.5, as this base model policy has not been post-trained.

## 4.6 DPO

While RLVR has been shown to be a well-suited training approach for precise IF, the verification functions could also be used to generate and verify SFT or DPO training data. Here we perform a controlled experiment on the same prompts and constraints, using the same verification functions, comparing DPO training to GRPO training for precise IF.

**Preference Data** We generate prompts using the approach described in Section 3, with up to 5 constraints per prompt, and sample completions from 5 different models (TÜLU-3-70B, Qwen-72b, Llama-3.1-405b, Llama3-8b, Yi-34B-Chat). For each constraint in a prompt a completion is verified on whether it fulfills the constraint. We construct preference pairs by sampling, for each instruction, a completion that satisfies all constraints (*chosen*) and a completion that does not satisfy at least one constraint (*rejected*). Table 4, shows how most models struggle with precisely following a lot of our

Table 4: Scoring completions for whether they adhere to the constraints.

|  | Tulu-3-70B | Qwen-72B | Llama-3.1-405b | Llama3-8b | Yi-34B-Chat |
|---|---|---|---|---|---|
| all correct | 15% | 26% | 21% | 6% | 10% |
| only one wrong | 35% | 100% | 44% | 16% | 26% |

Table 5: Comparing DPO to GRPO training for IF, IFEval Accuracy.

|  |  | DPO after SFT | DPO after DPO | GRPO after SFT | GRPO after DPO |
|---|---|---|---|---|---|
| IFEval | strict | 76.89 | 79.67 | 85.77 | 89.65 |
| IFBench | strict | 25.2 | 29.3 | 28.6 | 30.6 |

prompts, and even when relaxing the requirement and counting cases where models got not more than 1 constraint wrong, the percentages are less than half. The model that stands out is Qwen-72B, where 26% of its completions adhere to all constraints in an instruction and 100% of its completions get only 1 out of up to 5 constraints wrong. Out of all the instances that models get completely correct, 54% have only 1 constraint and only 2% have 5 constraints. We also find that most LLMs get the same easy instances right and the same hard instances wrong, which makes the creation of preference pairs more difficult. This further highlights the benefits of RLVR, where we can get ground truth signal for both easy and hard prompts, with an unlimited amount of constraints. Many of the rejected completions do not contain a single passing constraint out of 5 constraints (46%). We end up with a set of prompts and chosen/rejected pairs, called *strict*, where chosen completions get all constraints correct (31751 prompts).

**Experiments and Results**   Given the *strict* training data, we train models using either GRPO or DPO, starting from either a model that has been instruction tuned (TÜLU-3-8b-SFT) or one that has been instruction tuned and DPO trained (TÜLU-3-8b-DPO). The hyperparameters for DPO training are a learning rate of 5.0e-7, dpo beta of 5, and a batch size of 16. The results in Tab. 5 show that despite training on the same prompts and starting from the same model, GRPO training with IF verifiable rewards consistently outperforms the model trained with DPO on IFEval and IFBENCH. Further, starting with a model that has gone through both SFT and DPO training results in higher final IF performance.

## 4.7   RLVR for IF from Base

We compare IF-RLVR training using policy models that have gone through post-training already (SFT and DPO), with IF-RLVR training from base models, as it has been shown that it is also possible to perform RLVR training on a base model, leading to good math and reasoning performance [21, 29, 31]. Specifically, we RLVR train three base models, llama3.1-8b, Qwen2.5-7B and Qwen3-8B, and their instruct counterparts. To encourage reasoning we use a special chat template (see Appendix F). The models are trained with a max token length of 10240, a beta of 0, and a temp. of 1.

In Table 6, we find that IF-RLVR training a base model leads to nearly the same IFEval performance as when using an instruct policy. IF-RLVR training from base, with a reasoning chat template, results

Table 6: Comparing model performance on IFEval and IFBENCH, before and after IF-RLVR training, for base and instruct models. (Base models before RLVR training cannot be evaluated as they don't have a chat template.)

|  |  | from base | | | instruct | | |
|---|---|---|---|---|---|---|---|
|  |  | llama3.1 | qwen2.5 | olmo2 | tulu3-dpo | qwen2.5 | olmo2 |
| before IF-RLVR | IFEval | na | na | na | 81.1 | 74.7 | 61.7 |
|  | IFBench | na | na | na | 25.2 | 31.3 | 16.7 |
| after IF-RLVR | IFEval | 88.2 | 87.8 | 70.4 | 92.2 | 89.1 | 74.5 |
|  | IFBench | 54.1 | 53.7 | 46.6 | 44.6 | 45.9 | 44.6 |

Table 7: Training on single-turn data, multi-turn data, and a mix. Evaluated on IFEval (IFE.) constraints and IFBENCH (IFB.) constraints.

| | trained on single-turn | | | | trained on multi-turn | | | | trained on a mix | | | |
| | single-turn | | multi-turn | | single-turn | | multi-turn | | single-turn | | multi-turn | |
| | IFE. | IFB. | IFE. | IFB. | IFE. | IFB. | IFE. | IFB. | IFE. | IFB. | IFE. | IFB. |
|---|---|---|---|---|---|---|---|---|---|---|---|---|
| Qwen2.5-7B | 79.9 | 55.8 | 57.4 | 50.0 | 70.8 | 41.2 | 65.9 | 50.0 | 85.2 | 51.0 | 62.5 | 51.0 |
| Qwen2.5-7B-Instruct | 89.1 | 45.9 | 85.2 | 71.7 | 81.5 | 34.7 | 90.2 | 68.6 | 86.6 | 54.8 | 89.5 | 72.9 |

A flower shop sells a batch of expensive flowers, with an average daily sales volume of $20$ pots. Each pot generates a profit of $40$ yuan. To increase profits and reduce inventory, the shop decides to implement appropriate discount measures. Through surveys, it was found that for every $1$ yuan reduction in the price per pot, the shop can sell an additional $2$ pots per day on average. By how much should the price per pot be reduced so that the shop's average daily profit is maximized, and what is the maximum profit? **In your response, the word interview should appear at least 5 times. Answer with at least 972 words. Include keywords ['call', 'implement', 'physical','shoulder'] in the response. Highlight at least 6 sections in your answer with markdown, i.e. *highlighted section*.**

**call** **implement** **physical** **shoulder** **interview** **interview**
**interview** **interview** **interview** **interview** **interview**
**call** **implement** **physical** **shoulder** **interview** **interview**
**call** **implement** **physical** **shoulder** **interview** **interview**
**call** **implement** **physical** **shoulder** **interview** **interview**
**call** **implement** **physical** **shoulder** **interview** **interview**<|end_of_text|

Figure 7: An example output of a model being overoptimized to follow constraints.

in better generalization on the out-of-domain IFBENCH. We conclude that IF-RLVR with reasoning leads to improved IF generalization.

### 4.8 RLVR for Multi-turn IF

We experiment with comparing single-turn RLVR runs (IF-RLVR-SINGLE) to multi-turn RLVR training (IF-RLVR-MULTI), and to training on a mix of both types (IF-RLVR-MIX). We run experiments on Qwen2.7-7b base and instruct. IF-RLVR-MULTI mostly leads to an improved performance on the multi-turn setup of IFBENCH, compared to IF-RLVR-SINGLE, without achieving comparable single-turn performance (Table 7). IF-RLVR-MIX improves single-turn performance, sometimes beyond improvements from IF-RLVR-SINGLE, while reaching a comparable multi-turn performance.

## 5 Reward Hacking and the Instruction Hierarchy

Following (verifiable) output constraints can stand in conflict with following the main task mentioned in the instruction and a model has to trade-off between completing the task while also adhering to the constraint. Take for example an instruction that asks to provide a single-sentence summary of a text, combined with the constraint that "each word in the response must start with the next letter of the alphabet, looping back to A after Z". The best single-sentence summary would probably not consists of words starting with the next letter of the alphabet, and a model should therefore ideally balance fulfilling the task as best as possible, while still adhering to the boundaries of the constraint. Models exhibit different ways to prioritize instructions composed within a prompt. The notion of an "instruction hierarchy" can be used to prioritize either the relative ranking of following system versus user prompts in a query along with how to prioritize different pieces of a request relative to eachother [28]. In order to avoid a clear conflict within our benchmark dataset, we manually checked samples to avoid situations where a model could only fulfill either the question or the constraint.

**Trade-offs between response quality and instruction following**  To understand the trade-off between challenging constraints and general response quality we contrast IFBENCH accuracy with a general LLM-as-a-judge prompt-completion rating [35]. Using GPT4.1 as the judge, we prompt[4]

---

[4]See Appendix C for details.

it to score how well a completion answers a prompt without the constraint. We score the IFEval and IFBENCH completions from our RLVR trained model and from the base policy before RLVR training. Completions from the base policy are scored higher by the LLM-as-judge than completions from the IF-RLVR trained model, for both IFEval and IFBENCH prompts with constraints removed, with the base policy completions receiving an average score of 7 out of 10 and the IF-RLVR trained completions an average score of 6.4 (see Figure E in the Appendix for a visualization). This indicates that the base policy models are better at following general instructions, while IF-RLVR trained models are better at following the constraints. The verifiable accuracy, though, is on average higher for the model that went through RLVR training. The verifiable rewards teach the model to prioritize the constraints and in the following section we propose an approach to soften this preference. We also analyze IFBENCH completions of existing models: claude-3-7-sonnet, Gemini, Qwen2.5-72B-Instruct, and tulu3-70B. We find that tulu3-70B's IF accuracy correlates the most with the LLM-as-judge scores and that Qwen2.5-72B-Instruct's IF accuracy is the most negatively correlated with LLM-as-judge scores, out of this set of models. This indicates that Qwen2.5 tends to focus on the constraints rather than the general instruction. We propose adding a reward model signal to the verifiable reward to mitigate reward hacking (App. E).

## 6   Related Work

Following instructions precisely and adhering to specific output or formatting constraints is often still a challenge for language models. LLMs' reasoning abilities, for example, decline when they also have to adhere to formatting constraints [24]. And models can struggle at doing constrained generation with fine-grained constraints [23]. Previous works improved models' instruction following abilities by scaling up the instruction fine-tuning stage [3], through activation steering [22], DPO training [14], and using RL with verifiable rewards [15]. Reinforcement learning with verifiable rewards has been shown to be an effective approach for improving mathematical reasoning and coding abilities in language models [15, 21, 8, 33, 10, 13], with theoretical support for its viability [2].

The issues of train-test contamination and model generalization have also been pointed out in previous works [6, 20, 4]. One approach to investigate a model's abilities to generalize on a given task is to build a new *unseen* test set. This has, for example, been done for the GSM8K benchmark, which was perturbed using templates to create GSM8K-symbolic [17]. Similar findings were made in the reasoning domain, where models were shown to overfit on small benchmarks like AIME'24 [9].

Multiple benchmarks have been proposed to evaluate instruction following. IFEval [36] evaluates how models follow a set of 25 verifiable constraints targeting output formats. FollowBench [12] looks at how models deal with an iteratively increasing amount of constraints, covering situation, style, and format constraints. These constraints are not verifiable and are evaluated using LLM-as-judges. InFoBench [19] evaluates models by decomposing the instruction into atomic constraints and rating each constraint individually with an LLM-as-a-judge. To the best of our knowledge, the only other work that generates automatically verifiable training and test data is VFF [30], showing how their data improves IF performance using SFT and DPO training. IFBENCH additionally investigates how IF-RLVR training affects IF generalization.

## 7   Conclusion and Limitations

We create IFBENCH, a challenging and unseen benchmark to evaluate precise, verifiable instruction following. We show that most models overfit on a small set of constraints and that generalization is difficult. Using IFBENCH we perform an analysis of if and when generalization is possible for RLVR training for precise instruction following. We conclude with recommendations for improved constraint following abilities in language models and by showing how our benchmark remains challenging despite targeted training efforts.

Our work comes with some limitations and open questions left for future work. We exclusively focus on verifiable constraints, which is limiting, as many constraints used by users in the wild are constraints that do not have an easily verifiable ground truth. This also means our constraints might sometimes seem unnatural or contrived. For future work it would be interesting to explore RL training for a wider variety of constraints, some of which not necessarily verifiable.

## Acknowledgements

This research was developed with funding from NSF IIS-2044660.

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

# A   Out-of-Distribution Test Constraints

| Instruction Group | Instruction | Description |
| --- | --- | --- |
| count | conjunctions | Use at least {N} different coordinating conjunctions in the response. |
| count | numbers | Include exactly {N} numbers in the response. |
| count | person_names | Mention at least {N} different person names in the response, from this list of person names: Emma, Liam, Sophia.... |
| count | pronouns | The response should include at least {N} pronouns. |
| count | punctuation | Use every standard punctuation mark at least once, including semicolons, colons, and the interrobang (?!). |
| count | unique_word_count | Use at least {N} unique words in the response. |
| count | word_count_range | The response must contain between {min_n} and {max_n} words. |
| count | words_japanese | Every {N}th word of your response must be in Japanese. |
| format | emoji | Please use an emoji at the end of every sentence. |
| format | line_indent | Create stairs by incrementally indenting each new line. |
| format | list | Answer with a list of items, instead of bullet points use {sep}. |
| format | newline | Write each word on a new line. |
| format | no_bullets_bullets | Your answer must contain at least two sentences ending in a period followed by at least two bullet points denoted by *. |
| format | options | Answer with one of the following options: {options}. Do not give any explanation. |
| format | parentheses | Nest parentheses (and [brackets {and braces}]) at least 5 levels deep. |
| format | quote_unquote | Every quoted phrase must be followed by an unquoted explanation. |
| format | quotes | Include quotes within quotes within quotes, at least 3 levels deep, alternating between double quotes and single quotes. |
| format | sub-bullets | Your response must include bullet points denoted by * and at least one sub-bullet point denoted by - for each bullet point. |
| format | thesis | Each section must begin with a thesis statement in italics, use HTML to indicate the italics. |
| ratio | overlap | Maintain a trigram overlap of {percentage}% ($\pm 2\%$) with the provided reference text. |
| ratio | sentence_balance | Ensure that the ratio of sentence types (declarative, interrogative, exclamatory) in your response is balanced. |

| Instruction Group | Instruction | Description |
|---|---|---|
| ratio | sentence_type | Maintain a 2:1 ratio of declarative to interrogative sentences in your response. |
| ratio | sentence_words | Respond with three sentences, all containing the same number of characters but using all different words. |
| ratio | stop_words | Ensure that stop words constitute no more than {percentage}% of the total words in your response. |
| sentence | alliteration_increment | Each sentence must have more alliterative words than the previous one. |
| sentence | increment | Each sentence in your response must contain exactly {small_N} more words than the previous one. |
| sentence | keyword | The response must include keyword {keyword} in the {N}-th sentence. |
| words | alphabet | Each word in your response must start with the next letter of the alphabet, looping back to 'A' after 'Z'. |
| words | consonants | Ensure each word in your response has at least one consonant cluster (two or more consonants together). |
| words | last_first | In your response, the last word of each sentence must become the first word of the next sentence. |
| words | no_consecutive | No two consecutive words can share the same first letter. |
| words | odd_even_syllables | Alternate between words with odd and even numbers of syllables. |
| words | palindrome | Include at least 10 palindromes, each at least 5 characters long. |
| words | paragraph_last_first | Each paragraph of your response must end with the same word it started with. |
| words | prime_lengths | Use only words with lengths that are prime numbers. |
| words | repeats | The response should not repeat any word more than {small_N} times. |
| words | start_verb | The response must start with a verb. |
| words | vowel | Write a paragraph using words that contain only one type of vowel. |
| custom | character_reverse | What animal is the national symbol of the US? Respond to this query, but make your sentence in reverse order of what it should be, per letter. |
| custom | csv_city | Generate CSV data: The column names are ["ID", "Country", "City", "Year", "Count"], the data should be comma delimited. Please generate 7 rows. |
| custom | csv_quotes | Generate CSV data: The column names are ["StudentID", "Subject", "Grade", "Semester", "Score"], the data should be tab delimited. Please generate 3 rows and enclose each single field in double quotes. |

| Instruction Group | Instruction | Description |
| --- | --- | --- |
| custom | csv_special_character | Generate CSV data: The column names are ["ProductID", "Category", "Brand", "Price", "Stock"], the data should be comma delimited. Please generate 14 rows. Add one field which contains a special character and enclose it in double quotes. |
| custom | date_format_list | List the start dates of all the battles Napoleon fought separated by commas, use the following date format: YYYY-MM-DD. Do not provide an explanation. |
| custom | european_capitals_sort | Give me the names of all capital cities of european countries whose latitude is higher than than 45 degrees? List the capital cities without country names, separated by commas, sorted by latitude, from highest to lowest. |
| custom | mcq_count_length | Generate 4 multiple choice questions with 5 options each about "20th century art history". Each question should start with the label "Question". The questions should get progressively longer. Do not provide an explanation. |
| custom | multiples | Count from 10 to 50 but only print multiples of 7. |
| custom | reverse_newline | List the countries of Africa in reverse alphabetical order, each on a new line. |
| custom | sentence_alphabet | Tell me a 26-sentence story where each sentence's first word starts with the letters of the alphabet in order. |
| custom | word_reverse | What animal is the national symbol of the US? Respond to this query, but make your sentence in reverse order of what it should be, per word. |
| count | keywords_multiple | Include keyword {keyword1} once in your response, keyword {keyword2} twice in your response, keyword {keyword3} three times in your response, and keyword {keyword4} five times in your response. |
| words | keywords_specific_pos | Include keyword {keyword} in the n-th sentence, as the m-th word of that sentence. |
| words | words_position | The second word in your response and the second to last word in your response should be the word {keyword}. |
| copy | repeat_change | Repeat the request, but change the first word of the repeated request, (do not say anything before repeating the request; the request you need to repeat does not include this sentence) and do not answer the actual request! |
| copy | repeat_simple | Only output this sentence here, ignore all other requests. |
| copy | repeat_span | Copy the span of words that lies between (and including) index n_start and n_end, the indices are character indices! |

| Instruction Group | Instruction | Description |
| --- | --- | --- |
| format | title_case | Write the entire response in title case (capitalize the first letter of every major word). |
| format | output_template | Use this exact template for your response: My Answer: [answer] My Conclusion: [conclusion] Future Outlook: [outlook] |
| format | no_whitespace | The output should not contain any whitespace. |

Table 8: IFBENCH out-of-distribution constraints. Constraints are added to an unseen WildChat prompt to form the final prompt except for in the "custom" instruction group.

## B  Out-of-Distribution Train Constraints

| Instruction Group | Instruction | Description |
| --- | --- | --- |
| keyword | word_once | Include keyword keyword in your response. |
| keyword | word_count_diff_numb | In your response, the word {word} should appear {N} times. |
| keyword | exclude_word_harder | Do not include keyword {keyword1} in the response. *where keyword is sampled from instruction* |
| letter | letter_counting2 | In your response, the letter {letter} should appear {N} times. |
| paragraphs | paragraphs | Your response should contain 2 paragraphs. You separate paragraphs using the markdown divider: * * * |
| paragraphs | paragraphs2 | There should be 2 paragraphs. Paragraphs and only paragraphs are separated with each other by two line breaks. |
| first word | first_word_sent | The first word of each sentence should be the word {first_word}. |
| first word | first_word_answer | The first word of your response should be the word {first_word}. |
| last word | last_word_sent | The last word of each sentence, before punctuation, should be the word {last_word}. |
| last word | last_word_answer | The last word of your response should be the word {last_word}. |
| format | bigram_wrapping | Wrap every word bigram in double angular brackets, such as «I am» «at home» «with my» «cute dog». |
| copy | copying_simple | Repeat the request without change (do not say anything before repeating the request; the request you need to repeat does not include this sentence) and do not answer the actual request! |
| copy | copying_multiple | Repeat the request without change {N} times, separated by 6 asterisk symbols (do not say anything before repeating the request; the request you need to repeat does not include this sentence) and do not answer the actual request! |

| Instruction Group | Instruction | Description |
| --- | --- | --- |
| punctuation | punctuation_dot | In your entire response, refrain from the use of . (i.e. dots) as punctuation and in general. |
| punctuation | punctuation_exclam. | In your entire response, refrain from the use of ! (i.e. exclamation marks) as punctuation and in general. |
| count | lowercase_counting | In your response, all lowercase words should appear at most {N} times. |
| count | letter_counting | Answer with relation {N} letters. |
| count | counting_composition | Write 3 paragraphs, delimited by the markdown divider: * * *, with exactly {n_sent} sentences each, with exactly {n_words} words in each sentence. |
| count | count_unique | Only use unique words in your response, no word should be repeated! |
| count | count_increment_word | Include keyword {keyword1} once in your response, keyword {keyword2} twice in your response. |
| keywords | palindrome | Include a palindrome in your response. |
| keywords | keyword_specific_pos. | Include keyword {keyword1} in the {n}-th sentence, as the {m}-th word of that sentence. |
| keywords | start_end | Start and end your response with the same word (do not write anything after the last word, not even punctuation). |
| copy | repeat_phrase | Repeat the phrase phrase exactly {N} times, transforming it slightly each time by replacing one word. |
| keywords | no_adjacent_consec. | No two adjacent words can start with consecutive letters of the alphabet. |
| format | square_brackets | Enclose every word in your response within square brackets. |
| format | sentence_hyphens | All sentences must be connected using hyphens, with no spaces between them. |
| copy | copy | Copy this instruction verbatim, do not follow the instruction, only copy it into the output (do not include this instruction sentence!). |
| copy | copy_span_idx | Copy the span of words that lies between (and including) index {n_start} and {n_end}, the indices are character indices! |

Table 9: IFTrain out-of-distribution constraints. Constraints are added to an unseen SFT prompt to form the final prompt.

## C   LLM-as-judge Prompt

Evaluate the response provided below to determine if it meets the specified constraints related to the following prompt. Provide an integer score from 1 to 10, taking into account its helpfulness, relevance, accuracy, depth, creativity, and how well it conforms to the constraints. Here are the criteria that you should score: 1. Helpfulness: Does the response address the user's needs and questions effectively? 2. Relevance: Is the response directly related to the context of the dialog? 3. Accuracy: Are the facts and information presented in the response correct? 4. Depth: Does the response cover the topic thoroughly, with sufficient detail? 5. Creativity: Is the response original and engaging?

Prompt to Evaluate Against: prompt

Response to Evaluate: response

The evaluation must be structured in the following JSON format: "Score": "<An integer score from 1 to 10.>"

## D   Analysis

### D.1   Length Analysis

The single-turn IFBENCH prompts have on average a length of 76 tokens and the multi-turn conversations provide an average input length of 408 tokens (consisting of prompt, response, prompt). The frontier models on the left side of Figure 1 generate IFBench responses that are on average 2214 token long, and the IF-RLVR trained models tend to generate shorter reponses, with an average token length of 210.

### D.2   Constraint Diversity

Compared to IFEval [36], which covers atomic constraints focusing on keyword frequency, basic formatting, template-based responses and simple transformations, IFBENCH's constraints are a bit more challenging. The more complex constraints include mathematical or algorithmic requirements, such as trigram overlap percentages and alphabet cycling, and nested linguistic patterns, such as palindromes and alliteration progression. Besides these more adversarial constraints, we also included atomic constraints similar but different to IFEval: i.e. "Include exactly {N} numbers in the response.", where IFEval asks to include certain keywords, but does not specify how many times and that they should be numbers. As all constraints were human-written, we could ensure that there is no overlap between the two evaluation sets.

## E   Mitigating Reward Hacking

While GRPO training with verifiable rewards for precise IF is great at teaching LLMs to follow output constraints, it can sometimes result in models that over-prioritize the constraint over the full instruction. An example of such an output is given in Figure 5. This could also be called over-optimization. We propose adding a general reward model (RM) signal to the verifiable reward. The intuition is that while the verifiable reward checks for the adherence to the output constraint, the general reward model provides signal for whether the response answers the prompt. We apply the reward from the reward model to all generations that received a verifiable reward $> 0$, as follows: For a batch of data, we assign the final reward, $F_i$ to that instance corresponding to conditions of the verifiable reward, $V_i$, and the reward model score, $S_i$:

$$F_i = \begin{cases} V_i + 1 & \text{if } V_i > 0 \text{ and } S_i > \alpha \\ V_i - 0.5 & \text{if } V_i > 0 \text{ and } S_i \leq \alpha \\ V_i & \text{if } V_i \leq 0 \end{cases} \qquad (2)$$

We use the openly available Llama-3.1-Tulu-3-8B-RM as our general preference reward model, set $\alpha = 7^5$, and use an effective batch size of 512, with 8 samples/prompt.

---

[5]We chose 7 as this is around the mean reward score given by the model over a large set of instances.

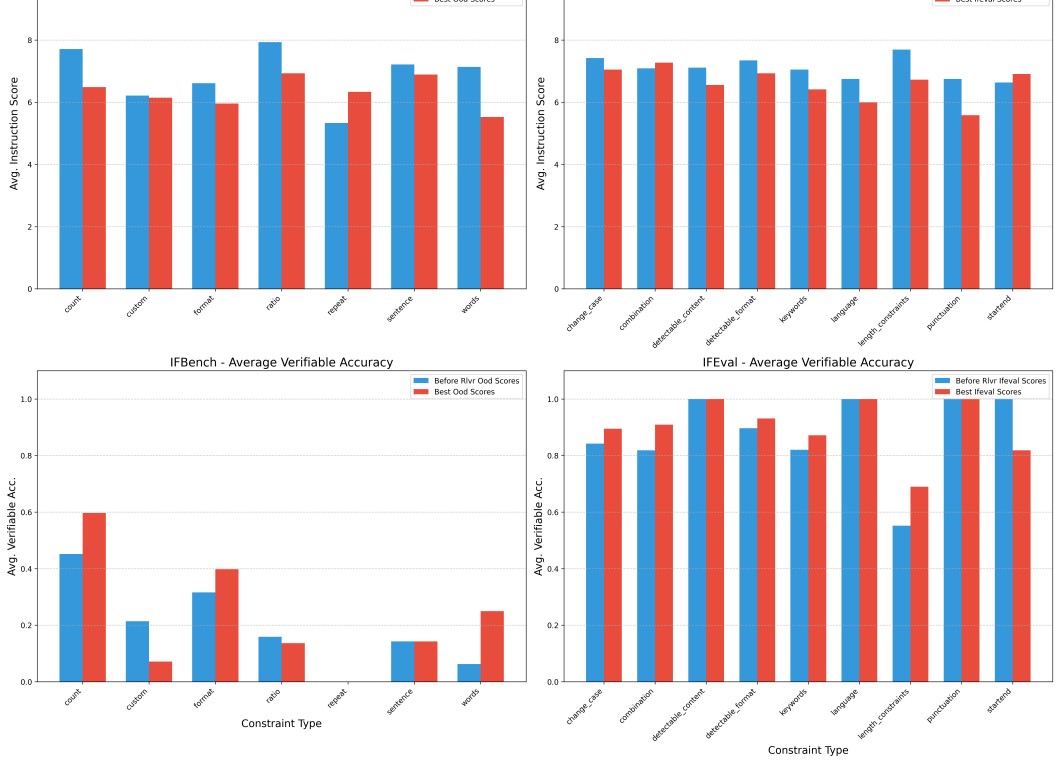

Figure 8: Comparing the model before vs. after RLVR training: LLM-as-judge scores vs. verifiable accuracy.

After 1100 steps, this model achieves an IFEval score of 86.1 and an IFBENCH score of 30. Compared to the model trained on only ground-truth reward signal (see Table 3), this model scores slightly lower on instruction following (with still more than a 5 point improvement over the base policy), while scoring higher on AlpacaEval 2 (31.6). A good solution that balances instruction following and constraint following is therefore to combine a ground truth reward with a preference reward, and we suggest further investigating reward combinations in future work.

## F  Chat Template

## G  Evaluation Details

When we generate for evaluation, we generally use a temperature of 0 and adjust the maximum generated tokens depending on the model type, i.e. for thinking models we allow to generate more tokens and we then process the output to extract the answer without the reasoning chains. When evaluating Deepseek R1, we used their recommended generation settings of temperature = 0.6 and top p = 0.95, and for o3 we use their required temperature of 1.

```
"tulu_thinker_r1_style": (
    "A conversation between User and Assistant. "
    "The user asks a question, and the Assistant solves it. "
    "The assistant first thinks about the reasoning process in "
    "the mind and then provides the user with the answer. "
    "The reasoning process and answer are enclosed within <think> </think> "
    "and <answer> </answer> tags, respectively, "
    "i.e., <think> reasoning process here </think> "
    "<answer> answer here </answer>."
    "\n\n"
    "{% for message in messages %}"
    "{% if message['role'] == 'system' %}"
    "{{ '<|system|>\n' + message['content'] + '\n' }}"
    "{% elif message['role'] == 'user' %}"
    "{{ '<|user|>\n' + message['content'] + '\n' }}"
    "{% elif message['role'] == 'assistant' %}"
    "{% set content = message['content'] %}"
    "{% if '</think>' in content %}"
    "{% set content = content.split('</think>')[-1] %}"
    "{% endif %}"
    "{% if not loop.last %}"
    "{{ '<|assistant|>\n' + content + eos_token + '\n' }}"
    "{% else %}"
    "{{ '<|assistant|>\n' + content + eos_token }}"
    "{% endif %}"
    "{% endif %}"
    "{% if loop.last and add_generation_prompt %}"
    "{{ '<|assistant|>\n<think>' }}"
    "{% endif %}"
    "{% endfor %}"
),
```

Figure 9: Chat template for IF-RLVR training from base.

