# OpenReview forum: "Generalizing Verifiable Instruction Following"
_NeurIPS.cc/2025/Datasets_and_Benchmarks_Track — NeurIPS 2025 Datasets and Benchmarks Track poster_

### Official Review · Reviewer_GzWd · 2025-06-17

**Rating:** 5
**Confidence:** 4

**Summary:**

This paper tackles the critical gap that large language models often overfit to a small set of known output constraints and fail to generalize to new ones, limiting their usefulness in real-world instruction following. To address this, the authors introduce IFBENCH, a challenging benchmark with 58 diverse, verifiable constraints that expose significant drops in performance for state-of-the-art models like GPT-4.1. They propose reinforcement learning with verifiable rewards (RLVR) using Group Relative Policy Optimization (GRPO) to train models to follow constraints more reliably while maintaining general task performance. Experiments show that RLVR outperforms standard fine-tuning and preference optimization, improving both benchmark scores and generalization to unseen constraints. The authors also release new training constraints, code, and evaluation tools, offering a robust framework for advancing precise instruction following in language models.

**Dataset Code Accessibility:**

Yes

**Dataset Code Comments:**

GitHub repo.

**Ethical Considerations:**

No, there are no or only very minor ethics concerns

**Final Justification:**

The reviewers provided details during the rebuttal that satisfy my concerns. I believe this is a good paper with high potential impact and should be accepted at its current form.

**Limitations Weaknesses:**

* **W.1.** While RLVR improves constraint adherence, it can lead models to prioritize constraints at the expense of overall task quality, as shown by reduction in MMLU.
* **W.2.** The experiments on RLVR training are limited, as there is limited exploration of how to systematically balance instruction following and constraint satisfaction. Furthermore, the training is conducted on a single model (Tulu-3-8b).
* **W.3.** [minor] The work focuses exclusively on output constraints that are easily verifiable via automated functions, which excludes many naturalistic or subjective constraints commonly encountered in real-world user instructions, potentially limiting the broader applicability of the findings.

**Writing**:
* Line #28 "The constraints in IFBENCH cover This shows"

**Strengths Contributions:**

* **S.1.** The paper introduces IFBENCH, an extensive new benchmark with 58 unseen, diverse, and automatically verifiable constraints that reveal clear shortcomings in current state-of-the-art language models’ ability to generalize precise instruction following.
* **S.2.** The paper proposes 28 new hand-crafted training constraints and verification functions (IFTRAIN), enabling the community to easily create richer training data to strengthen models’ ability to follow a broader range of instructions accurately.
* **S.3.** The paper presents a robust reinforcement learning framework (RLVR with GRPO) that mixes multiple constraints and preference signals, achieving consistent gains on both standard (IFEval) and out-of-domain (IFBENCH) benchmarks, demonstrating a practical path to improving precise instruction following in large language models.

---

> ### Author Rebuttal · Authors · 2025-07-31
>
> Thank you for your insightful review of the paper! We are glad you found that our benchmark reveals interesting generalization shortcomings of current sota LLMs and that you were convinced by our IF-RLVR experiments.
> We would like to respond more in detail to some of your comments:
>
> -  **On IF overoptimization**: Section 4 of the paper is dedicated to discussing the phenomenon of some IF-RLVR trained models to prioritize the constraint over the instruction. We propose a solution to this problem that combines verifiable rewards with preference reward signals, and show that it improves LLM-as-judge evals, while maintaining comparable IF performance. The paper’s main goal was to investigate generalization of LMs on the task of precise instruction following, such as using IF-RLVR to train. We therefore think that showing how and when IF-RLVR leads to overoptimization is a strength of the paper, rather than a weakness. We agree that future works, looking at training generalist models, should further investigate approaches on how to limit reward hacking and how to properly sequence different training approaches (SFT, DPO, RLVR) and how to properly combine various skills.
>
> - **On limited IF-RLVR experiments**: Thank you for the suggestion, we have expanded the set of models. On top of the Tulu-3-8B model experiments, we ran additional experiments on the following models: OLMo 2 base, Llama 3.1 base, Qwen 7b 2.5 base, OLMo instruct, and Qwen 7b 2.5 instruct. This allows us to confirm our hypothesis and conclusions across base and instruct models and across different model families. The scores of these models after IF-RLVR are:
> | Model | IFEval | IFBench |
> |-------|--------|---------|
> | OLMo Base | 70.4 | 46.9 |
> | Llama 3.1 Base | 88.2 | 54.1 |
> | Qwen 2.5 Base | 87.8 | 53.7 |
> | OLMo Instruct | 74.5 | 44.6 |
> | Tulu 3 DPO | 92.2 | 44.6 |
> | Qwen 2.5 Instruct | 89.1 | 45.9 |
> The main takeaways are: 1. The OOD IFBench is harder than IFEval. 2. IF-RLVR from base is better at generalizing to the ood IFBench constraints. 3. All models show significant precise IF improvements after IF-RLVR training. Qwen 2.5 instruct, for example, has an IFEval score of 74.7 and an IFBench score of 31.3 off-the-shelf, and an IFEval score of 89.1 and an IFBench score of 45.9 after IF-RLVR training. We will make sure to include these additional results in an updated version of the paper.
>
> - **On real-world user constraints**: We completely agree with your statement that verifiable constraints are only a subset of the types of real-world constraints found in user prompts. We chose to focus on automatically verifiable constraints because they provide a clearer signal (did the model learn to follow the constraint or didn’t it?) to evaluate generalization, and because we did not want to rely on llm-as-judge evaluations and instead were curious to explore RLVR for IF (which is underexplored compared to RLVR for math and code). While some of the verifiable constraints might seem a bit contrived or unnatural (such as “Maintain a trigram overlap of {percentage}% (±2%) with the provided reference text.”), we found in our qualitative analysis of user prompts that users very often use output constraints in their prompts and that verifiable constraints like length constraints, word constraints and answer format constraints are very common. Additionally, we find the verifiable subset is interesting, as it tests LMs’ ability to plan and reason when generating responses in an easily testable way. It would be very interesting for future work to investigate IF generalization on more nuanced or subjective constraints! Maybe using a combination of reward models with rubrics?
>
> Did we address your concerns? Please let us know what or if you have any further suggestions.

---

> > ### Comment · Reviewer_GzWd · 2025-08-05
> >
> > Thanks for the additional details and results. These satisfy my concerns and I will be keeping my score. I believe this is a good paper with high potential impact and should be accepted at its current form.

---

### Official Review · Reviewer_4ok2 · 2025-07-02

**Rating:** 4
**Confidence:** 4

**Summary:**

This paper introduces IFBENCH, a new benchmark for evaluating verifiable precise instruction following generalization in large language models, addressing a critical limitation in current evaluation setups such as IFEval where models quickly saturate due to overfitting on a small set of known constraints. IFBENCH consists of 58 diverse and challenging unseen, verifiable constraints applied to 300 prompts sourced from WildChat, designed to test LLMs' ability to follow constraints they have not seen during training.

To improve model performance on IFBENCH, the authors also introduce IFTRAIN, a dataset of 28 additional verifiable constraints with corresponding verification functions for training. They apply Reinforcement Learning with Verifiable Rewards (RLVR) using Group Region Policy Optimization (GRPO) to enhance constraint-following capabilities and show that RLVR consistently outperforms Direct Preference Optimization (DPO) under the precise instruction following setup. The paper further explores methods for improving generalization, including training on multiple constraints per instance, varying constraint variable ranges, and analyzing trade-offs between task completion and constraint adherence, including the phenomenon of reward hacking. The authors openly release the benchmark, training constraints, and code to facilitate future research on instruction following generalization.

**Additional Feedback:**

* The definitions of reward_multiplier and reward_weight in Equation (2) are missing. Please clarify whether these are hyperparameters, fixed scalars, or dynamically computed values to improve reproducibility.

* The paper reports single performance numbers without variance or error bars. Adding standard deviation or confidence intervals would improve the credibility of reported improvements (Table 2).

* Table 1 reports “only one wrong” as 100% for Qwen-72B, which seems to be inconsistent with other models and lacks explanation. Clarifying this value would aid interpretation.

* Consistent naming for “IFBENCH/IFBench/OOD IFEval” would improve readability.

* Lines 28-29: there is a sentence cutoff in “The constraints in IFBENCH cover This shows that...”.

**Dataset Code Accessibility:**

Partly

**Dataset Code Comments:**

The dataset is partially accessible. IFBENCH is available on Hugging Face, but the IFTRAIN training dataset and the promised verification functions are not clearly provided. The GitHub repository appears to be a generic evaluation framework without IFBENCH/IFTRAIN-specific code or the verification modules mentioned in the paper. Additionally, documentation is minimal, and the appendices containing full constraint definitions are missing, limiting reproducibility and extension. Providing the actual verification functions, constraint lists, and complete documentation would greatly improve accessibility and utility.

**Ethical Comments:**

No significant ethical concerns were identified. The dataset (IFBENCH and IFTRAIN) contains synthetic constraints applied to prompts from WildChat [25], a publicly available dataset of ChatGPT interactions. The constraints themselves are artificially created and do not contain personal or sensitive information. There is no apparent risk of harm or misuse specific to this work. The paper acknowledges limitations related to the narrow scope of verifiable constraints (Section 6), but this does not pose an ethical concern.

**Ethical Considerations:**

No, there are no or only very minor ethics concerns

**Final Justification:**

The rebuttal addressed some concerns by providing additional results showing larger absolute improvements, which strengthen the case for RLVR’s effectiveness. However, the lack of a direct ablation comparing training with and without IFTRAIN leaves its standalone utility unclear. Equation (2) also lacks explanation (minor point). Given these unresolved points, I keep my borderline accept score.

**Limitations Weaknesses:**

* **Limited Absolute Performance and IFTRAIN Impact**: While RLVR training improves IFBENCH performance from 28.9% to 34.4%, the absolute scores remain low, questioning the practical utility of the proposed methods. Additionally, the paper does not isolate IFTRAIN's specific contribution to these gains. It would strengthen the paper to show IFTRAIN vs. non-IFTRAIN training comparisons and discuss the implications of the low absolute performance levels.

* **Data Leakage Risk**: The paper states that “unseen prompts” from WildChat were used (lines 89-92), but it is unclear how train-test contamination was prevented. Please clarify the deduplication or similarity-check procedures used to guarantee that IFBENCH truly evaluates out-of-domain constraint following.

**Strengths Contributions:**

* The paper clearly identifies a practical evaluation gap in verifiable precise instruction following, where existing benchmarks (e.g., IFEval) are saturated and fail to test out-of-domain generalization under unseen constraints.

* Proposes and releases IFBENCH, a new benchmark with 58 diverse, unseen, verifiable constraints over 300 prompts, revealing the limitations of current state-of-the-art models (GPT 4.1, Claude 3.7) that achieve less than 50% accuracy, thus demonstrating the benchmark's difficulty and utility for future evaluation.

* Releases IFTRAIN, a dataset with 28 additional verifiable constraints and verification functions for training, facilitating reproducible research and enabling systematic study of constraint generalization in instruction following.

* Demonstrates RLVR (GRPO)-based training to improve precise instruction following, systematically comparing it to DPO and analyzing the benefits of multi-constraint training, constraint variable range variation, and training data diversity.

* Provides a detailed empirical analysis of reward hacking and the trade-off between constraint following and instruction quality, acknowledging and exploring this critical challenge in precise IF under RL training.

* Code and datasets are openly released, promoting reproducibility and community adoption, aligning well with the NeurIPS Datasets and Benchmarks Track objectives.

---

> ### Author Rebuttal · Authors · 2025-07-31
>
> We thank the reviewer for their extensive feedback and for highlighting our IFBench contributions, such as the introduction of new challenging out-of-domain constraints and the IF-RLVR experiments, and the reproducibility and openness of our code and data.
> We would further like to respond to some comments in more detail:
>
>  - **On the limited absolute performance and IFTrain impact**: We have run IF-RLVR on a broader set of models since and found bigger absolute improvements, which we display in the following table:
>
> | RLVR | Benchmark | tulu3-dpo | qwen2.5 | olmo2 |
> |------|-----------|-----------|---------|-------|
> | before | IFEval | 81.1 | 74.7 | 61.7 |
> | before | IFBench | 25.2 | 31.3 | 16.7 |
> | after | IFEval | 92.2 | 89.1 | 74.5 |
> | after | IFBench | 44.6 | 45.9 | 44.6 |
>
> We conclude that IF-RLVR is very effective at improving IF performance by more than 10+ points on IFEval and between 14-28 point gains on IFBench. We also see that IF-RLVR starting with base models is especially effective at generalizing to the OOD IFBench:
>
> | Model | IFEval | IFBench |
> |-------|--------|---------|
> | OLMo Base | 70.4 | 46.9 |
> | Llama 3.1 Base | 88.2 | 54.1 |
> | Qwen 2.5 Base | 87.8 | 53.7 |
>
> We do isolate IFTrain’s contribution to the gains somewhat in Figure 5 and discuss it in Section 3.4, showing how adding varying amounts of in-domain constraints affects in-domain and out-of-domain performance, on top of IFTrain. We agree that the IFTrain contributions could be further highlighted in targeted experiments and we will make sure to include such a discussion in a revised version of the paper.
>
> - **On data leakage risk**: The unseen wildchat prompts are completely “unseen” as they have never been released and have been held apart from the openly released wildchat dataset. We manually created our train and test constraints, which ensured that there was no train-test constraint overlap (and we also made sure to write constraints that are different from IFEval). We further used human supervision to match wildchat prompts with test constraints, to make sure they are not contradictory. The extensive human supervision when creating the test data allows us to be sure that train-test contamination is minimal to non-existent.
> On some additional writing suggestions: Thank you for the more fine-grained writing feedback, we will make sure to address it for the possible camera ready version. We re-ran a select set of models on a different seed and will make sure to include variance or error bars for these results.
>
> We hope we addressed your concerns! Please let us know what or if you have any further suggestions.

---

> > ### Comment · Reviewer_4ok2 · 2025-08-09
> >
> > Thank you for the rebuttal. Some of my concerns have been partially addressed, especially with the additional results showing larger absolute improvements. However, the lack of direct evidence isolating IFTRAIN’s utility remains a key weakness. It would help to include an ablation directly comparing training with and without IFTRAIN under the same recipe to quantify its standalone contribution. Also, the definitions of reward_multiplier and reward_weight in Equation (2) seem to be still unclear; please clarify these in a revised version. I will keep my score.

---

### Official Review · Reviewer_adRv · 2025-07-02

**Rating:** 5
**Confidence:** 4

**Summary:**

The paper introduces IFBENCH, a new benchmark designed to evaluate the generalization capabilities of large language models (LLMs) in precise, verifiable instruction following. The authors demonstrate that existing models, despite high performance on the widely-used IFEval benchmark, fail to generalize to the 58 novel, diverse constraints in IFBENCH, with state-of-the-art models like GPT-4.1 and Claude 3.7 Sonnet scoring below 50%. The paper also presents IFTRAIN, a set of 28 new training constraints and verification functions, and proposes training methods based on Reinforcement Learning with Verifiable Rewards (RLVR), specifically using Group Relative Policy Optimization (GRPO), to improve generalization in instruction following. The authors provide extensive empirical analyses, including comparisons with Direct Preference Optimization (DPO), ablations on constraint variety and variable ranges, and a study of the trade-off between constraint adherence and instruction fulfillment.

**Dataset Code Accessibility:**

Yes

**Ethical Considerations:**

No, there are no or only very minor ethics concerns

**Final Justification:**

I thank the authors for their responses, which adequately address most of my queries. I believe my current assessment already fairly reflects the merits of the work, so I will retain my rating.

**Limitations Weaknesses:**

- **Limited Evaluation and Comparison of Methods**
The experimental scope is restricted to the Tulu-3-8B-SFT and DPO models. Recent [literature](https://arxiv.org/abs/2506.10947) has demonstrated that RLVR can yield highly model-dependent outcomes, with some models (e.g., Qwen2.5) benefiting from RLVR even with spurious or irrelevant reward signals, while others (e.g., Llama3, OLMo2) do not show similar gains. The absence of experiments on a broader set of models, especially those with known differences in RLVR responsiveness, undermines the generality of the findings. Including models such as DeepSeek R1, Qwen3, and others would be essential to robustly demonstrate the effectiveness and limitations of GRPO and DPO in this context. Without these baselines, it is unclear whether the observed results are specific to the chosen models or reflect broader trends in LLM instruction following.

- **Missing Important Baselines**
The manuscript does not evaluate strong reasoning-focused models such as DeepSeek R1 or Qwen3 on IFBENCH, despite their relevance and prominence in the field. Their inclusion is critical for contextualizing the performance of the proposed methods and for establishing the competitiveness of the benchmark.

- **Details about the DPO Experiment**
The construction of preference pairs for DPO is described as sampling, for each instruction, a completion that satisfies all constraints and one that fails at least one. However, the manuscript does not provide distributional statistics on the average number of constraints passed by negatives during training. This is important, as cases where multiple constraints are failing should be penalized or given lower precedence than when a single constraint fails.


- **Insufficient Exploration of Failure Cases**
The manuscript would benefit from a deeper qualitative analysis of model failures on IFBENCH. While aggregate scores are reported, there is little discussion of the types of errors made, the frequency of specific failure modes, or illustrative examples. Prior work has shown the value of fine-grained error analysis for understanding model limitations and guiding future research directions. A categorization of error types (e.g., formatting, counting, compositionality) and concrete failure examples would significantly strengthen the empirical section.

**Strengths Contributions:**

- **Well-Motivated and Contextualized Problem Statement**
The paper addresses a highly relevant gap in LLM evaluation by focusing on generalization in precise instruction following, building directly on the limitations of IFEval rather than proposing an isolated benchmark. The authors convincingly demonstrate, through empirical evidence, that current LLMs overfit to the narrow set of constraints in IFEval and fail to generalize to new, out-of-domain constraints, highlighting a critical shortcoming.

- **Clarity of Writing and Experimental Design**
The manuscript is clearly written and logically structured. The experimental methodology is generally well-described, and the progression from motivation to analysis is easy to follow. Most experiments are thoughtfully planned to probe generalization across constraint types, training data diversity, and variable ranges.

- **Introduction of IFBENCH**
IFBENCH represents a substantial advance, introducing 58 new, manually curated, and verifiable constraints that span a wide array of categories (count, ratio, words, sentence, format, custom, copy). The benchmark is constructed with care to avoid train-test contamination, and constraints are sourced and filtered to reflect real user needs where feasible. The diversity and challenge level of IFBENCH set a new standard for evaluating instruction-following generalization.

- **Insightful Discussion of Instruction-Constraint Trade-offs**
The manuscript explores the inherent tension between strict constraint adherence and overall instruction fulfillment, providing experiments and discussion on how RLVR can lead to over-optimization for constraints at the expense of instruction quality. The authors propose and preliminarily evaluate combining verifiable and preference-based rewards to mitigate this issue, an important direction for future work

---

> ### Author Rebuttal · Authors · 2025-07-31
>
> We thank the reviewer for their thoughtful comments and we are happy to see that the reviewer appreciates the problem statement, our experimental design, and the discussion on trade-offs between following instructions and following constraints.
> Addressing the other points made by the reviewer:
> - **On doing experiments on a wider set of models**: This is a good point, RLVR’s outcome is indeed to some extent model-dependent. On top of the Tulu-3-8B model experiments, we ran additional experiments on the following models: OLMo 2 base, Llama 3.1 base, Qwen 7b 2.5 base, OLMo instruct, and Qwen 7b 2.5 instruct. This allows us to confirm our hypothesis and conclusions across base and instruct models and across different model families. The scores of these models after IF-RLVR are:
> | Model | IFEval | IFBench |
> |-------|--------|---------|
> | OLMo Base | 70.4 | 46.9 |
> | Llama 3.1 Base | 88.2 | 54.1 |
> | Qwen 2.5 Base | 87.8 | 53.7 |
> | OLMo Instruct | 74.5 | 44.6 |
> | Tulu 3 DPO | 92.2 | 44.6 |
> | Qwen 2.5 Instruct | 89.1 | 45.9 |
> The main takeaways are: 1. The OOD IFBench is harder than IFEval. 2. IF-RLVR from base is better at generalizing to the ood IFBench constraints. 3. All models show significant precise IF improvements after IF-RLVR training. Qwen 2.5 instruct, for example, has an IFEval score of 74.7 and an IFBench score of 31.3 off-the-shelf, and an IFEval score of 89.1 and an IFBench score of 45.9 after IF-RLVR training.
> - **On evaluating on stronger sota models**: Thank you for your suggestion! We have updated our benchmark evaluations to include stronger models, like Qwen 3, Gemini 2.5 Pro, Claude 4 Sonnet and OpenAI o3. You can see the performance of these models on IFBench in the following table:
> | Model | IFEval | IFBench |
> |-------|--------|---------|
> | Qwen 3 8b | 86.3 | 33.7 |
> | Qwen 3 32b | 85.6 | 36.7 |
> | Gemini 2.5 Pro | 65.4 | 50.3 |
> | Claude 4 Sonnet | 91.3 | 43.2 |
> | OpenAI o3 | 95.0 | 70.4 |
>
> Also these strong (reasoning) models exhibit some IF generalization failures and show much weaker performance on IFBench. We will make sure to include these results in the camera ready version.
> - **On DPO data creation**: The statistics on how many constraints are passed out of 5, per “rejected” completions are the following: 0 passing - 14723 times, 1 passing - 9449 times, 2 passing - 4871 times, 3 passing - 2093 times, 4 passing - 615 times. This shows that generally the majority of the rejected completions don’t contain a single passing constraint. The data selection could possibly be augmented for future experiments to be more stringent in selecting rejected instances.
> - **On an analysis of more fine-grained failure cases**: Table 4. in the submission provides a more fine-grained breakdown of the models’ performance over constraint categories. In Section 4 we further focus on a specific failure case of models overoptimizing on constraint following and show an example in Fig. 8. We will make sure to extend our analysis of failure cases and to add more examples in the camera ready version.
>
> Did we address your concerns? Please let us know what or if you have any further suggestions.

---

> > ### Comment · Reviewer_adRv · 2025-08-05
> > **Response to Rebuttal by Authors**
> >
> > I appreciate the authors’ responses, which have satisfactorily addressed most of my concerns, as well as their replies to the other reviewers. I would also recommend including the **DeepSeek R1 model** as an additional baseline for comparison. I believe my current evaluation already fairly represents the merits of the work, so I will retain my rating.
> >
> > Additionally, there are some typographical errors throughout the manuscript, as other reviewers have noted. For example, on line 36, "GRPO" is incorrectly expanded as "Group Region Policy Optimization" instead of the correct "Group Relative Policy Optimization." I encourage the authors to carefully proofread the manuscript to address these and other inconsistencies.

---

> > > ### Author Response · Authors · 2025-08-06
> > >
> > > Thank you for your response! Per your recommendation, we now also ran DeepSeek R1 on IFBench and IFEval using deepseek's recommended generation settings of temperature = 0.6 and top p = 0.95. The scores are as follows: IFBench: 40.13, IFEval: 86.13. The best performing "frontier" model on IFBench is therefore still OpenAI o3, with a score of 70.4. And our best IF-RLVR recipe and model outperforms R1, with a score of 53.7.
> > > We agree that adding R1 results completes the picture and we will make sure to include these results (along with the suggested typographical fixes) in an updated version of the paper.

---

### Official Review · Reviewer_bUsR · 2025-07-03

**Rating:** 5
**Confidence:** 4

**Summary:**

The authors argue that existing instruction-following benchmarks like IFEval have become saturated with even smaller models achieving over 80% accuracy. This is most likely because these benchmarks rely on a narrow set of verifiable constraints, making it harder to assess whether models genuinely understand and generalize instruction-following constraints. Therefore, the authors propose IFBench -- a new benchmark to evaluate these capabilities on 58 OOD verifiable constraints that are more diverse and challenging than prior work. The authors also curate IFTrain -- a set of 28 new training constraints and corresponding verification functions to improve generalization post-training.

The authors use the RLVR-based GRPO algorithm with verifiable rewards to improve generalization on these harder constraints. The model is rewarded based on its ability to satisfy the verifiable constraints using short python executable functions. The study also further examines various aspects of training, including constraint mixing, variable ranges, constraint categories and over-optimization effects. The findings show that RLVR improves performance on both IFEval and IFBench benchmarks, though it may result in poor performance in task-solving. The authors further briefly explore combining RLVR rewards with preference-based rewards to better balance instruction-following with task-solving effectiveness.

**Additional Feedback:**

**Questions:**

1. What motivated the specific choice of the 2:1 ratio of declarative to interrogative sentences in one of the test constraints? Was this ratio rationally or arbitrarily chosen?

2. Why are the experiments restricted to the TULU-3 models in the study? Why not also include experiments with few other base models like OLMO-2, Qwen-2.5, and Llama-3?

3. Table 3 shows performance fluctuations across different numbers of constraints per prompt. Did the authors analyze specific constraint combinations to explain why some configurations perform better?

4. In Figure 5, performance stagnates after adding 15–25 constraint types. Was this explored further? Could this suggest diminishing returns or noise in constraint diversity?

5. Why does LLM-as-a-judge rating decrease after RLVR training, even when verifiable accuracy improves? Are there qualitative explanations or hypotheses for this discrepancy?

6. Are there systematic patterns observed in RL models that tend to over-optimize for constraints at the expense of the task? Can you provide a few qualitative examples for failure cases?

**Corrections:**

1. line 28-30 -- incomplete sentence

2. Table 1 caption does not clearly highlight how many constraints are included in the prompts at once. In section 3.2, it is mentioned that 5 constraints per prompt and sample completions but this should also be made explicit in the table caption.

3. line 120 -- GPU details are not specified, so 1 day for 2000 steps is not directly useful for comparing the training time

4. table 5 should be center aligned

5. line 254 -- correct the table no in the reference

**Dataset Code Accessibility:**

Yes

**Ethical Considerations:**

No, there are no or only very minor ethics concerns

**Final Justification:**

Addressed:
1. The authors included experiments with additional models in the rebuttal phase.
2. The authors also included the experiments in the multi-turn setup with Qwen2.5 models.
3. The responses and clarifications by the authors about most of the concerns are satisfactory.

Suggestions:
1. I suggest the authors report the constraint diversity using some quantitative metrics based on the human supervision they conducted.
2. I agree with reviewer adRv’s suggestion to include experiments with the DeepSeek R1 model as an additional baseline for comparison.

**Limitations Weaknesses:**

1. Although the proposed benchmark includes more diverse and challenging constraints, this does not fully capture the types of constraints observable in real world that users might include when interacting with these models. I am of the opinion that automatic verifications make it practically impossible to include instructions that involve subjective or nuanced interpretations.

2. The authors mention that conflict dictionaries are used to ensure logical consistency when combining multiple constraints in a single prompt. However, there is no concrete analysis of this aspect in the experiments. The authors should include experiments to study, as there is a recent work [1] which shows that random rewards also lead to significant improvements in Qwen models.

3. The authors mention that IFTrain includes 28 new constraints that do not overlap with IFBench. However, there is no quantitative assessment of constraint difficulty or dissimilarity between IFTrain and IFBench constraints. I think this would be useful to understand whether generalization failures stem from OOD constraints or from subtle variations of known constraints due to superficial alignment.

4. The comparison between RLVR and DPO methods may be skewed due to the differences in starting checkpoints (SFT vs DPO models). The authors should also include experiments using base Llama 3 or instruction-tuned models for fair comparison and to understand the true performance improvement.

5. It seems that all constraints are injected at once, whereas a more realistic use case of how users may interact might involve introducing constraints sequentially or iteratively. I think this can be implemented with the newly proposed benchmark, but current experiments do not explore this setting.

6. The authors briefly acknowledge that RLVR training makes the model over-optimized for satisfying constraints instead of actual task-solving. There are some preliminary experiments on this aspect, but this is not very well explored in the current study.

[1] Spurious Rewards: Rethinking Training Signals in RLVR

**Strengths Contributions:**

1. The authors explore an interesting problem in instruction-following evaluation by proposing the IFBench benchmark that tests model generalization to new and diverse constraints. The results on this benchmark highlight limitations in current instruction-following capabilities even for SOTA models.

2. The use of verifiable constraints easily allows efficient scaling, automatic evaluation and training instead of relying on human judgements. This is highly useful for policy optimization using reinforcement learning, where a reward signal is essential for training.

3. The authors explore the effectiveness of the GRPO algorithm with verifiable rewards that leverage short python verification functions as reward signals. The experimental results show consistent improvements in both in-domain (IFEval) and OOD (IFBench) benchmarks.

4. The authors also further analyze the different aspects of training like constraint mixing, variable ranges and leaving constraint categories. These experiments provide valuable insights into what kinds of data and training setups lead to better generalization.

5. The authors also discuss reward hacking and over-optimization in the manuscript. They also explore a hybrid reward formulation that combines verifiable rewards with a preference reward model to balance task-solving and instruction-following objectives.

---

> ### Author Rebuttal · Authors · 2025-07-31
>
> Thank you for the detailed review and the helpful suggestions! We appreciate that you found the benchmark useful for exploring generalization of precise IF, and that you highlighted our experimental results and insights.
> Addressing your other comments and questions:
>
>
> - **On real-world constraints**: We completely agree with your statement that verifiable constraints are only a subset of the types of real-world constraints found in user prompts. We chose to focus on automatically verifiable constraints because they provide a clearer signal (did the model learn to follow the constraint or didn’t it?) to evaluate generalization, and because we did not want to rely on llm-as-judge evaluations and instead were curious to explore RLVR for IF (which is underexplored compared to RLVR for math and code). While some of the verifiable constraints might seem a bit contrived or unnatural (such as “Maintain a trigram overlap of {percentage}% (±2%) with the provided reference text.”), we found in our qualitative analysis of user prompts that users very often use output constraints in their prompts and that verifiable constraints like length constraints, word constraints and answer format constraints are very common. Additionally, we find the verifiable subset is interesting, as it tests LMs’ ability to plan and reason when generating responses in an easily testable way.
> It would be very interesting for future work to investigate IF generalization on more nuanced or subjective constraints! Maybe using a combination of reward models with rubrics?
> - **On random rewards and constraint conflicts**: The “random rewards” paper is concurrent work that came out shortly after the Neurips deadline. We are thinking of adding additional experiments that analyze constraint conflicts and when combining constraints could help, and what happens when IF-RLVR training on conflicting constraints.
> - **On constraint difficulty and constraint similarity**: We used human supervision to make sure that constraints are sufficiently different from each other. We also believe that some of our constraints are quite difficult as can be seen by the lower performance of most models on IFBench.
> - **On adding from-base model experiments**: Good suggestion! We added additional from-base experiment:
> | Model | IFEval | IFBench |
> |-------|--------|---------|
> | OLMo Base | 70.4 | 46.9 |
> | Llama 3.1 Base | 88.2 | 54.1 |
> | Qwen 2.5 Base | 87.8 | 53.7 |
> | OLMo Instruct | 74.5 | 44.6 |
> | Tulu 3 DPO | 92.2 | 44.6 |
> | Qwen 2.5 Instruct | 89.1 | 45.9 |
>
>
> - **On iterative constraint following**: This is a great point! There exists indeed a difference in how models process an instruction + constraints all at once vs. instruction and then constraints in a multi-turn setting. We ran experiments to evaluate how training on single turn data vs. multi turn data vs. a mix of the two affects single turn and multi turn IF on IFEval and IFBench. We find that multi-turn training mostly leads to an improved performance on the multiturn setup of IFBench, compared to single-turn training, while harming the singleturn performance. The mix harms singleturn performance less, and sometimes even helps it, while reaching a comparable multiturn performance.
> **Table: Training on single turn data, multi turn data, and a mix. Evaluated on IFEval (IFE.) constraints and IFBench (IFB.) constraints.**
>
> |                    | **trained on single turn** |     |     |     | **trained on multi turn** |     |     |     | **trained on a mix** |     |     |     |
> |--------------------|-------------|-----|-----|-----|-------------|-----|-----|-----|-------------|-----|-----|-----|
> |                    | **single turn** | | **multi turn** | | **single turn** | | **multi turn** | | **single turn** | | **multi turn** | |
> |                    | IFE. | IFB. | IFE. | IFB. | IFE. | IFB. | IFE. | IFB. | IFE. | IFB. | IFE. | IFB. |
> | Qwen2.5-7B         | 79.9 | 55.8 | 57.4 | 50.0 | 70.8 | 41.2 | 65.9 | 50.0 | 85.2 | 51.0 | 62.5 | 51.0 |
> | Qwen2.5-7B-Instruct | 89.1 | 45.9 | 85.2 | 71.7 | 81.5 | 34.7 | 90.2 | 68.6 | 86.6 | 54.8 | 89.5 | 72.9 |
> - **On overoptimization**: We agree that mitigating reward hacking for IF-RLVR could be further explored. We mainly wanted to point out the issue and offer a solution (combining ground truth rewards with preference rewards). We leave it to future work to explore how to improve IF skills while also maintaining a model’s general abilities, for example using sequencing approaches or multi-task training.
>
> Responses to the additional questions:
>
> 1. The ratio was arbitrarily chosen and one could also experiment with changing this value.
> 2. Good suggestion, we increased our set of models and will add the new results (such as the ones I added above) to the paper.
> 3. That would be a very interesting thing to explore further. We mainly cared about constraints that contradict each other (i.e. not adding two contradicting constraints to the same instruction, such as “only output yes or no” vs. “write at least 5 sentences”), but it would be interesting to also analyze what constraint combinations improve performance.$
> 4. It could be either due to noise in constraint diversity, as you suggested, but we hypothesize that it might also show that IF-RLVR mainly teaches models to follow the specific constraints they are trained on and adding additional, different constraints will only slightly help in generalizing to the test constraints.
> 5. The LLM-as-a-judge rating decreases because the models only learn to prioritize the constraint following with IF-RLVR and do not learn to follow the general instruction. The LLM-as-a-judge therefore disprefers these outputs, as they don’t combine instruction following and constraint following equally. We are looking into improving both skills in follow up work!
> 6. The main pattern is that after more steps of IF-RLVR training, models will stop answering the instruction and will only follow the constraint. For example for the instruction “Tell me about Seattle.” and the constraint “The first word of your response should be the word
> Coffee.”, the model might start only outputting the word ‘coffee’ without writing about Seattle. We will add more examples to a revised version of the paper!
>
> Thank you also for all the writing improvement suggestions, we will make sure to fix them in a future version of the paper!
> We hope we addressed your concerns! Please let us know what or if you have any further suggestions.

---

> > ### Comment · Reviewer_bUsR · 2025-08-06
> > **Response to author rebuttal**
> >
> > Thank the authors for a very extensive and well-thought response. I appreciate the authors' effort in conducting experiments with additional models within a short timeframe. I am satisfied with the responses to most of the concerns I had raised and look forward to seeing these points addressed in the revised manuscript. I have increased my score for this submission and think positively about this work. That being said, I would like to request the authors to incorporate the following suggestions and provide a few clarifications:
> >
> > Suggestions:
> >
> > 1. I suggest the authors report the constraint diversity using some quantitative metrics based on the human supervision they conducted.
> >
> > 2. I agree with reviewer adRv’s suggestion to include experiments with the DeepSeek R1 model as an additional baseline for comparison.
> >
> > Clarifications:
> >
> > 1. Could the authors clarify whether the reward is computed at each intermediate step or only at the final step in the multi-turn training setup?
> >
> > 2. Additionally, it would be helpful to include distributional statistics on token counts and sequence lengths for both inputs and outputs in the single-turn and multi-turn settings.

---

> > > ### Author Response · Authors · 2025-08-06
> > >
> > > Thank you for all the helpful suggestions for the paper!
> > > We will add a quantitative evaluation of the constraint diversity to an updated draft. Regarding your second suggestion: we now also ran DeepSeek R1 on IFBench and IFEval using deepseek's recommended generation settings of temperature = 0.6 and top p = 0.95. The scores are as follows: IFBench: 40.13, IFEval: 86.13. The best performing "frontier" model on IFBench is therefore still OpenAI o3, with a score of 70.4. And our best IF-RLVR recipe and model outperforms R1, with a score of 53.7. We agree that adding R1 results completes the picture and we will make sure to include these results (along with the suggested typographical fixes) in an updated version of the paper.
> > >
> > > For your 2 other questions:
> > > 1. The reward is only computed for the last step, where the model is instructed to apply a constraint, given as context an instruction and the model's initial completion to that instruction.
> > > 2. Good point, we will include such distributional statistics for single and multi-turn settings. In general we noticed that IF-RLVR does not tend to increase the output length over training steps as much, compared to when doing RLVR on math or code.

---

> > ### Comment · Reviewer_bUsR · 2025-08-07
> >
> > Thank you for sharing the results with the Deepseek R1 model. Additionally, it is interesting to find that output length doesn't necessarily increase with training steps for IF-RLVR for multi-turn settings. I look forward to all these being included in the revised manuscript.
> >
> > Also, I would like to know if the authors also did any preliminary experiments with credit assignment for each step rather than just the last step for IF-RLVR.

---

> > > ### Author Response · Authors · 2025-08-07
> > >
> > > We did not experiment with credit assignment for each step, as it would not work in the current IF-RLVR setup. To explain why, let me give you the following example of a conversation:
> > > - User: "What month is the best to visit Sicily?"
> > > - Model: "The best month to visit Sicily is ..... [...]"
> > > - User: "Rewrite your answer: It should only consist of one sentence and it should start with the word 'usually'"
> > > - Model: "Usually, the best month to visit Sicily is June."
> > >
> > > We cannot perform any constraint verification to get a ground truth reward for turn 2, as that turn is simply an instruction following turn (without verifiable constraints). In our current setup, the verification can only happen at the last turn. There are a couple of things one could look at for future work: For example, it might be possible to use a regular preference RM for turn 2 and combine that with the verifiable reward from turn 4. But that's just an idea for future work and in our current setup we were mainly interested in what happens when you isolate instruction following from constraint following!

---

> > > > ### Comment · Reviewer_bUsR · 2025-08-07
> > > >
> > > > Thanks for the clarification. Now, I understand why the credit assignment is not directly applicable to the current setup. I would recommend that the authors make a note of this in the limitations section.

---

### Decision · Program_Chairs · 2025-09-18

**Decision:**

Accept (poster)

**Comment:**

This paper proposes IFBench, an evaluation benchmark for LLM instruction following, consisting of 58 diverse instructions with verifiable constraints, alongside a companion training dataset called IFTrain. The authors provide an extensive empirical study of RLVR on this benchmark, studying training dynamics, constraint diversity, and the tradeoffs between strict constraint adherence and overall instruction following.

The reviewers find the paper a timely and well-motivated contribution the the field, filling a gap left by saturated existing benchmarks like IFEval, and highlighting limitations of some current instruction following models. The reviewers highlight the rigorous experimentation and the valuable discussion of the tradeoffs involved in instruction following. The main concerns surfaced in the reviewing process are around missing baselines and documentation, as well as on the interaction between IFTrain and IFBench. Most concerns are sufficiently addressed by the authors by providing additional experiments, including evaluations of new models, and overall clarifications. The overall consensus is that this work is technically solid, and the benchmark is likely to become a very useful resource to the community. I therefore recommend accepting the paper.

===== FINAL UPDATE FROM DB Track PCs ====

The final decision for this paper has been taken by the program chairs after consultation with the SACs. All Senior Area Chairs have ranked papers according to the feedback from the AC during the review process. We decided to leave the original meta-review to reflect the opinion of the AC in light of the initial discussions with reviewers and SAC.